# Investigation of the Entry Pathway and Molecular Nature of σ1 Receptor Ligands

**DOI:** 10.3390/ijms24076367

**Published:** 2023-03-28

**Authors:** Gianmarco Pascarella, Lorenzo Antonelli, Daniele Narzi, Theo Battista, Annarita Fiorillo, Gianni Colotti, Leonardo Guidoni, Veronica Morea, Andrea Ilari

**Affiliations:** 1Institute of Molecular Biology and Pathology (IBPM), National Research Council of Italy (CNR), 00185 Rome, Italyandrea.ilari@cnr.it (A.I.); 2Department of Biochemical Sciences “A. Rossi Fanelli”, Sapienza University of Rome, 00185 Roma, Italy; 3Department of Physical and Chemical Sciences, University of L’Aquila, 67100 L’Aquila, Italy; 4Protein Production Facility, Structural Biology Laboratory, Elettra Sincrotrone Trieste, 34149 Basovizza, Italy

**Keywords:** σ1 receptor, Huntington’s disease, molecular dynamics, virtual screening, fluorescence titration

## Abstract

The σ1 receptor (σ1-R) is an enigmatic endoplasmic reticulum resident transmembrane protein implicated in a variety of central nervous system disorders and whose agonists have neuroprotective activity. In spite of σ1-R’s physio-pathological and pharmacological importance, two of the most important features required to fully understand σ1-R function, namely the receptor endogenous ligand(s) and the molecular mechanism of ligand access to the binding site, have not yet been unequivocally determined. In this work, we performed molecular dynamics (MD) simulations to help clarify the potential route of access of ligand(s) to the σ1-R binding site, on which discordant results had been reported in the literature. Further, we combined computational and experimental procedures (i.e., virtual screening (VS), electron density map fitting and fluorescence titration experiments) to provide indications about the nature of σ1-R endogenous ligand(s). Our MD simulations on human σ1-R suggested that ligands access the binding site through a cavity that opens on the protein surface in contact with the membrane, in agreement with previous experimental studies on σ1-R from *Xenopus laevis*. Additionally, steroids were found to be among the preferred σ1-R ligands predicted by VS, and 16,17-didehydroprogesterone was shown by fluorescence titration to bind human σ1-R, with significantly higher affinity than the prototypic σ1-R ligand pridopidine in the same essay. These results support the hypothesis that steroids are among the most important physiological σ1-R ligands.

## 1. Introduction

Neurodegenerative diseases with distinct genetic etiologies and pathological phenotypes appear to share common mechanisms of neuronal cellular dysfunction, including excitotoxicity, calcium dysregulation, oxidative damage, endoplasmic reticulum (ER) stress and mitochondrial dysfunction. Glial cells, including microglia and astrocytes, play an increasingly recognized role in both the promotion and prevention of neurodegeneration. Sigma receptors, particularly the σ1 receptor (σ1-R) subtype, are a unique class of intracellular proteins, expressed in both neurons and glia of multiple regions within the central nervous system (CNS), which can modulate many biological mechanisms associated with neurodegeneration.

The human σ1-R, encoded by the *SIGMAR1* gene, is an enigmatic ER-resident transmembrane protein implicated in a variety of diseases affecting the CNS, such as depression, drug addiction and neuropathic pain [1]. This receptor is the focus of intense research since σ1-R ligands endowed with agonistic activity have been shown to be neuroprotective [2]. Consistent with the neuroprotective role hypothesis, several σ1-R mutations have been shown to be associated with neurodegenerative diseases. Familial amyotrophic lateral sclerosis (ALS) patients have been reported to exhibit the missense mutation c.304G>C in the *SIGMAR1* gene. This mutation results in the glutamic acid to glutamine substitution at amino acid residue 102 of the encoded protein (p.E102Q). Expression of the E102Q mutant protein reduces mitochondrial ATP production, inhibits proteasome activity and causes mitochondrial injury [3] Additional connections to ALS and Huntington’s disease (HD) have emerged from studies of human genetics and mouse models [4,5,6].

Interestingly, σ1-R is an evolutionary isolate with no discernible similarity to other proteins. The only known σ1-R homologue is the yeast C-8 sterol isomerase ERG2. ERG2 is one of the proteins involved in the biosynthesis of ergosterol, which is essential to modulate fungal cell membrane fluidity, like cholesterol does in animal cells. In particular, ERG2 catalyzes the reaction that shifts the delta-8 double bond to delta-7 position in the B ring of sterols, thereby converting fecosterol to episterol [7].

Recent studies have allowed the 3D structure of σ1-R to be revealed, and residues involved in interactions with several ligands to be identified.

The 3D structures of the human σ1-R receptor (*Hs*σ1-R), in complex with two chemically divergent ligands, i.e., PD144418 (PDB ID: 5HK1) and 4-IBP (PDB ID: 5HK2), have been experimentally determined by X-ray crystallography [1,5] and are publicly available from the Protein Data Bank (PDB; URL: rcsb.org) [8]. While PD144418 is a σ1-R antagonist, the activity of 4-IBP has been reported to differ from that of classic agonists or antagonists [9]. In both of these structures, *Hs*σ1-R is a trimer whose constituent monomers are related to one another by a three-fold symmetry axis. Each *Hs*σ1-R protomer contains a 30-residue long transmembrane α-helix, at the N-terminus, and a 189-residue long C-terminal domain in the aqueous phase, adjacent to the membrane. The trimeric interface, which has a surface of ~9300 Å^2^, is formed by the C-terminal domains of adjacent monomer pairs, whereas the three N-terminal transmembrane helices are located at each corner of the trimer and are involved in lattice contacts. Subsequent biochemical studies have demonstrated that σ1-R is a type II membrane protein, i.e., its orientation is such that the short N-terminal tail is located in the cytoplasm and the C-terminal domain is in the ER lumen [10] From a structural point of view, the C-terminal domain assumes a cupin-like β-barrel fold, with the ligand at its center, flanked by four alpha helices. In both structures, the ligand binding site is completely buried within the protein, thereby offering no information about the ligand access route to the binding site. Nevertheless, two possible points of access have been proposed: the first (pathway 1) through the protein surface in contact with the aqueous medium, at the level of polar residues Gln135, Glu158 and His154; and the second (pathway 2) through the protein surface in contact with the membrane, at the level of the two C-terminal helices α4 and α5.

In a subsequent study by the same group, three additional *Hs*σ1-R structures were reported, in complex with three different ligands: one agonist, (+)-pentazocine (PDB ID: 6DK1), and two antagonists, haloperidol (PDB ID: 6DJZ) and NE-100 (PDB ID: 6DK0). In addition, the mechanism of access to the binding site was investigated by molecular dynamics (MD) simulations [5]. In spite of the fact that the five ligands complexed with *Hs*σ1-R have different chemical structures, the overall conformation of the C-terminal ligand binding domain is conserved in all of the *Hs*σ1-R protomers present in the five structure determinations. Indeed, the root mean square deviation (RMSD) values calculated after optimal superposition of Cα atoms is in the range 0.11–0.6 Å for all 15 protomers [11]. Even if the ligand binding site is occluded in the latest three structures as well, in the (+)-pentazocine-bound structure the α4 helix shifts away from the α5 helix with respect to the position observed in all the other complexes [1,5]. This conformational difference has been ascribed to the fact that (+)-pentazocine has a non-linear structure, which would clash with the α4 helix if this assumed the same conformation. as in the complexes with the other, linear, compounds (i.e., PD144418, haloperidol, NE-100 or 4-IBP).

Accelerated MD simulations (aMD) of the *Hs*σ1-R monomer inserted in a hydrated lipid bilayer were performed in four different conditions [5]: (i) on the (+)-pentazocine-bound structure (PDB ID: 6DK1); (ii) on the haloperidol-bound structure (PDB ID: 6DJZ); (iii) in condition (i) except for the absence of the ligand; and (iv) in condition (i) with the ligand placed in water at a distance > 10 Å from the protein. As a result of these simulations, it was proposed that major conformational rearrangements of the protein should take place to make the binding site accessible to the ligand. The first of these rearrangements is the opening of the ‘lid’ of the β-barrel, following disruption of the backbone hydrogen bonds between W136 in β-strand 6 and A161 in β-strand 9. This lid, comprising β-strands 6 and 7 and the loop that connects them, points toward the aqueous medium and is the region of the β-barrel farthest from the ligand. Then, the backbone hydrogen bonds between E123 and R175 are broken and the β-strands 5 and 10, where these residues are located, respectively, separate from each other, thus exposing the binding pocket. Finally, the ligand enters the binding site and assumes a position similar (i.e., RMSD < 3.0 Å) to that observed in the crystal structure [1,5].

The route of ligand access to the *Hs*σ1-R binding site has been further investigated by both additional MD simulations [12] and experimental studies [13].

Steered MD simulations were performed on the monomer of PD144418-bound *Hs*σ1-R, which is the highest resolution (2.5 Å) *Hs*σ1-R structure (PDB ID: 5HK1) [1,5], either in the presence or in the absence of ligand [12]. An external force was applied to induce opening of the binding pocket and detachment of the PD144418 ligand from the *Hs*σ1-R monomer, since no significant changes in ligand binding cavity dimensions were observed following a 435 ns, standard MD simulation. Both proposed pathways were investigated: (i) pathway 1, directed towards the aqueous solvent, through a polar region occluded by Q135, H154 and E158 in β-strands 6 and 8 and in the loop between β-strands 8 and 9, respectively; and (ii) pathway 2, directed towards the membrane, through the α-helices 4 and 5 that are in contact with the membrane. Based on the magnitude of the force and the time required for the ligand to be completely dissociated from the protein, the pathway connecting the ligand binding site with the aqueous milieu was proposed to be the most likely ligand access route, in agreement with the results of the previously performed MD studies [1,5]. This hypothesis implies that, to reach the binding site, the ligand would initially interact with the polar residues Q135, H154 and E158 even if, based on lipid/water partition coefficients, the three compounds studied in this work would preferentially be associated with lipid environments [12].

In contrast with the results of MD simulations, the results of a recent work performed on the σ1-R homologue from *Xenopus laevis* (*Xl*σ1-R) indicate that the ligand is more likely to enter the binding site from the membrane side (pathway 2), thanks to conformational changes determining an opening between the α4 and α5 helices, rather than from the aqueous medium (pathway 1), following major structural rearrangements of the cupin-fold domain [13]. In this work, seven *Xl*σ1-R structures were solved by X-ray crystallography, and differ from one another in: (i) state of ligation (i.e., they are either in the free state or in complex with one of the known ligands, the PRE084 agonist or the S1RA antagonist); (ii) ligand binding site conformation (i.e., closed vs. open-like conformation); and (iii) presence of mutated residues. The *Xl*σ1-R protomer has high sequence and structure conservation with *Hs*σ1-R, except for the orientation of the transmembrane α1 helix. The trimeric arrangement is also conserved, although 12 or 24 *Xl*σ1-R protomers (i.e., four or eight trimers) are present in the asymmetric unit. The first evidence supporting ligand entrance from the membrane side is that, in the multimeric arrangement observed in the closed conformation (PDB ID: 7W2B), the trimers are packed with one another in such a way that the lid region of the cupin domain (that comprises W133, equivalent to W136 of *Hs*σ1-R) of one of the three constituent monomers is buried by two monomers belonging to different trimers. This arrangement is expected to prevent the conformational rearrangements proposed to take place to allow ligand entrance by MD simulations [5,12]. Soaking of either PRE084 or S1RA into this structure led to new structures (PDB ID: 7W2C and 7W2D, respectively), where each protomer comprises a ligand within the binding site, in the absence of significant conformational changes at the protomer, trimer or whole dodecamer level. This result indicates that the ligands are unlikely to have accessed the binding site through a drastic rearrangement of the cupin domain in the tightly packed crystals and more likely to have entered from the membrane side. The second evidence supporting pathway 2 is that, in the open-like conformation (PDB ID: 7W2E), there is an opening between the α4 and the α5 helices, contributed by a conformational change of the Y203 (equivalent to Y206 in *Hs*σ1-R) side-chain, which is large enough to allow the passage of ligands such as PRE084 and S1RA. Conversely, the rest of the structure is very similar to the closed conformation, both at the protomer and trimer level, and differs only in the relative orientation of the α1 transmembrane helix with respect to the C-terminal ligand binding domain. Following either co-crystallization or soaking with PRE084, the ligand was found within the binding site of every protomer (PDB ID: 7W2G and 7W2F, respectively), in the absence of changes of the rest of the structure, except for the conformation of the Y203 side-chain. The third, and most compelling, evidence is that blockage of the α4-α5 helices opening led to a substantial reduction in the fraction of *Xl*σ1-R binding sites occupied by PRE084, as measured by isothermal titration calorimetry (ITC). This blockage was achieved by first mutating residues L179 (equivalent to L182 in *Hs*σ1-R) and Y203, which are on opposite sides of the opening, with cysteine residues, then by either modifying C179 and C203 with a bulky reagent, or catalyzing the formation of a disulfide bond between C179 and C203 by oxidation. In the latter case, an increase in *Xl*σ1-R binding sites occupied by PRE084 was partially reverted by re-reduction of the C179-C203 disulfide bond. The *Xl*σ1-R C179/C203 double mutant structure in complex with S1RA (PDB ID: 7W2H) is very similar to all other closed or open-like conformations, at both the protomer and trimer level, and the ligand is bound in a similar way, demonstrating that these mutations do not significantly perturb either protein structure or ligand binding activity. Interestingly, although the structures in coordinate files 7W2B (“closed” conformation) and 7W2E (“open-like” conformation) were solved in the putative apo-form, an electron density peak was identified in proximity to the *Xl*σ1-R binding site in the Fo-Fc electron density map of both structures [13].

In addition to the preferred pathway of ligand access to, and exit from, the binding site, one of the important open questions about σ1-R is the nature of the physiological ligand(s). A remarkable σ1-R feature is the high chemical structure diversity of its ligands, some of which have other receptors as their main targets [14].

Several pharmacophores have been proposed for σ1-R ligands, sharing the following features: (i) one positively ionizable group, which some models specify to be a basic amine group that acts a hydrogen bond acceptor; (ii) several hydrophobic regions, with variations in distances and angles, generally including aromatic rings; and (iii) in some of the models, one additional polar group, possibly including one oxygen atom [15,16,17,18,19,20,21]. Indeed, in all ligand-bound *Hs*σ1-R and *Xl*σ1-R structures, a basic nitrogen of the ligand establishes a charge–charge interaction with the highly conserved E172 and E169, respectively, a residue that has been demonstrated by mutagenesis experiments to be essential for ligand binding [22]. A second *Hs*σ1-R acidic residue, D126, which had been demonstrated to be essential as well, forms a hydrogen bond with E172, indicating that it exists in a protonated state when ligands are bound. Only a few other, not conserved, polar interactions are observed, which involve hydroxyl or ether oxygens of the ligand, and variable *Hs*σ1-R or *Xl*σ1-R main-chain carboxyl oxygens or side-chain hydroxyl groups. With the exception of Y103, which engages in an aromatic stacking interaction in both structures, all of the other σ1-R-ligand interactions involve the hydrophobic residues lining the binding pocket (i.e., V84, W89, M93, L95, L105, F107, I124, W164 and L182) and hydrophobic regions of the bound ligands [1,5]. In spite of the extensive conservation of these pharmacophoric features, other known σ1-R ligands, such as several neurosteroids (e.g., the dehydroepiandrosterone and pregnenolone agonists and the progesterone antagonist) [23], do not comprise either of the minimal pharmacophore regions, leaving the question about the chemical structure of the physiological σ1-R ligand(s) wide open.

In this work, we tried to contribute to the elucidation of these two essential open questions about σ1-R function, i.e., physiological ligand identity and route of access to the binding site. To this end, we performed MD simulations on *Hs*σ1-R trimeric assembly, embedded in a physiological-like lipid bilayer, in the absence of any ligand and without the application of any bias, to try and reconcile the results reported in previous studies [1,5,12,13] about the entry pathway of the ligand. Additionally, to shed light on the nature of the endogenous *Hs*σ1-R ligand(s), we used a combination of computational virtual screening (VS), electron density maps fitting of selected compounds resulting from VS, and implementation and application of a fluorescence titration assay to measure ligand binding to *Hs*σ1-R in vitro.

Our MD results highlight conformational changes involving the α4 helix and the beta strands β4, β5 and β10, thereby supporting the hypothesis of ligand entrance from the membrane side (pathway 2). VS procedures and electron density map fitting indicated that steroid-based compounds are preferred endogenous σ1-R ligands, and one of them, 16,17-didehydroprogesterone, was shown, by fluorescence titration, to bind *Hs*σ1-R in vitro with higher affinity than pridopidine and iloperidone.

Taken together, the results obtained in this work support the hypothesis that steroid-based compounds are favored endogenous σ1-R ligands and that they access the σ1-R binding site from the protein side in contact with the membrane.

## 2. Results

### 2.1. Molecular Dynamics

The dynamics of the apo form of the trimeric *Hs*σ1-R, embedded into a bilayer resembling the membrane composition of the ER, as shown in Figure 1, was investigated by means of all-atoms MD simulations for 1.5 µs. Along the simulated trajectory, the RMSD of the backbone atoms of the trimer calculated with respect to the initial atomic positions increased to about 0.6 nm, while the RMSD of the single monomers in the last 500 ns fluctuated between 0.3 and 0.5 nm (see Appendix A). Overall, the secondary structures of the three monomers were found to remain rather stable along the simulation time, as confirmed by the time evolution of the number of H-bonds calculated within the single monomers (see Appendix A). Notably, the root mean square fluctuation of protein residues averaged over the last 500 ns of simulation showed different values for the three monomers, with the average fluctuations of monomer B larger than monomer C, and fluctuations of monomer C larger than monomer A (see Appendix A). Apart from the different fluctuations involving cytosolic protein loops, possibly due to a limited sampling time, significant differences between monomer B and the other monomers were found for residues 115–128 (in strands β4 and β5), and residues 172–188 (in strand β10 and helix α4). Looking at the simulated structures, we found that these two regions underwent a significant conformational change in monomer B. In Figure 2, the distance between the α4-helix (residues 180–188) and the coil between strands β4 and β5 (residues 118–121, shown in yellow and grey in panels B–C, respectively), both of which rest on the lipid bilayer, was monitored along the simulated trajectory, and a significant spacing between these two regions was observed in the last 150 ns of simulation. This conformational change also involved the β5-strand (residues 123–125, partially unfolded at the end of the simulation) and the β10-strand (residues 172–175), resulting into the opening of the substrate cavity (see Figure 2). Intriguingly, we found a possible correlation between such a conformational change and the breaking/formation of salt bridges between three residues, namely R175, E102 and E123, located in strands β10, β3 and β5, respectively (see Figure 3). By monitoring the distances between R175 and the two residues E102 and E123 (see Figure 3), we found three different behaviors for the three monomers. In monomer A, R175 formed an almost permanent salt bridge with E123. The same occurred in monomer B up to 1 µs of simulation; subsequently, the salt bridge between R175 and E123 was lost and a new salt bridge between R175 and E102 was formed. In monomer C, for the first 600 ns the behavior was similar to the behavior found in monomer A, and then both the R175/E123 and R175/E102 salt bridges were lost. The breakage of the salt bridge between R175 and R123 occurring in monomer B after 1 µs is likely to affect the subsequent spacing between the β5 and β10 strands, to which E123 and R175 belong, with the consequent opening of the cavity. In this context, E102 in the β3 strand may play a crucial role in triggering the opening of the substrate cavity, by forming a salt bridge with Arg175 and, therefore, inducing the breaking of the salt bridge between R175 and E123.

### 2.2. Virtual Screening against Hsσ1-R Structures and ERG2 Molecular Model

To try and identify common structures among *H*σ1-R ligands, we performed VS on a 21,359-compounds dataset. This dataset comprises: all human metabolites; several categories of steroid-based compounds (i.e., sterol lipids, sterols, steroids, androgens, estrogens and compounds belonging to the cholesterol or ergosterol biosynthetic pathway); known ligands of the σ1-R receptor and/or of the σ2-R receptor, which is known to share several ligands with σ1-R in spite of their different overall structure, as positive controls; and compounds approved for clinical use by the FDA or other regulatory agencies. This dataset comprises several compounds that are reported to bind human σ1R with very low affinity (i.e., Ki > 10,000 nM) by the Psychoactive Drug Screening Program (PDSP) Ki database [14], which we used as negative controls.

The VS was performed against two of the five available *Hs*σ1-R structures, namely the structures in coordinate files 5HK1, since it was solved with the highest resolution, and 6DK1_A, since it is the only one determined in complex with an agonist, rather than antagonist, compound, and against the molecular model of yeast ERG2 built by the AlphaFold2 program.

The results of these VS experiments are summarized in Table 1 for two subsets of hits, defined on the basis of the values of their receptor binding energy calculated by the program used for VS (E_calc_), namely: (i) the 20 hits with the lowest E_calc_; and (ii) all the hits whose E_calc_ does not differ more than 3 kcal/mol from the lowest E_calc_. The rationale for choosing the first set of hits is that it is commonly reported to be selected for detailed analyses in the literature, due to the fact that 20 is a small enough number of compounds for visual inspection. However, it has been reported that E_calc_ values have a standard deviation of 2–3 Kcal/mol [24]. It follows that hits whose E_calc_ differs by less than 3.0 Kcal/mol from the E_calc_ of the best hit may have an actual binding energy to the receptor similar to, or even better than, that of the best hit, and may, therefore, be all considered as “best hits”. For this reason, we chose to analyze in greater detail this second set of hits, which, from now on, will be referred to as “best-E3”. The results of the VS against the structures in coordinate files 5HK1 monomer A (5HK1_A) and 6DK1 monomer A (6DK1_A) for the hits in the “best-E3” subsets are reported in Appendix A, respectively, and plotted in Appendix A, respectively. We found that the “best-E3” subset for the VS against *Hs*σ1-R in coordinate files 5HK1 and 6DK1 comprise 1666 and 2987 hits, respectively, with E_calc_ between −13.10 and −10.10 kcal/mol and between −13.20 and −10.20 kcal/mol, respectively (Appendix A). Comparison of these results (Appendix A) shows that 1271 compounds are among the “best-E3” for both structures, whereas 395 and 1717 compounds are among the “best-E3” hits only for 5HK1 and 6DK1, respectively. As shown in Appendix A, there is no obvious correlation between the E_calc_ of the 1271 common “best-E3” towards 5HK1 and 6DK1. Given the high similarity between the *Hs*σ1-R structures in coordinate files 5HK1 and 6DK1 [11], these results indicate that VS results are significantly affected even by the very small side-chain variations induced upon *Hs*σ1-R binding by different ligands.

The full results of the VS against ERG2 molecular model for the “best-E3” subsets are reported in Appendix A and plotted in Appendix A. These hits fall into an E_calc_ range between −11.7 and −8.7 Kcal/mol, which is higher than those of the “best-E3” resulting from VS against the *Hs*σ1-R structure in coordinate sets 5HK1 and 6DK1, although the difference is not significant when the expected 2–3 kcal/mol standard deviation on E_calc_ values is taken into account [24]. Due to this expected standard deviation, the E_calc_ values for fecosterol and episterol, which are the substrate and product of the reaction catalyzed by the ERG2 protein, respectively, are higher (i.e., −10.3 and −9.4 kcal/mol, respectively) than the best hit (phaseolinisoflavan, E_calc_ = −11.7 kcal/mol), which does not contain a steroid nucleus; additionally, other compounds belonging to the ergosterol synthesis pathway and, therefore, likely to have structures able to bind ERG2, have an E_calc_ similar to, or higher than, that of unrelated compounds.

To verify whether the “best-E3” subsets were enriched with specific structures with respect to the whole 21,359 compounds dataset used for VS, we compared the number of compounds belonging to each category (e.g., metabolites, agonists/antagonists, etc.) comprised in this initial dataset with the number of hits of the same category comprised in the “best-E3” subsets, resulting from VS towards the *Hs*σ1-R structures in coordinate files 5HK1 and 6DK1, and towards the ERG2 molecular model (Table 2). Examination of these values shows that the “best-E3” subsets resulting from VS against the 5HK1 structure and ERG2 model are significantly enriched in compounds belonging to the agonists/antagonists category, which comprises experimentally validated σ1-R ligands, the ratio between the percentage of this category among the “best-E3” hits and among the starting set of compounds (R) being 2.8 for the 5HK1 structure and 2.4 for the ERG2 model. Conversely, no variation in the percentage of this category is observed in the results of VS against 6DK1. The percentage of compounds belonging to the “metabolites” category is significantly reduced among the “best-E3” subsets for both the *Hs*σ1-R structures and ERG2 model, with R values of 0.3, 0.3 and 0.7, respectively, whereas compounds approved by the FDA and other regulatory agencies do not show a regular trend (Table 2). Interestingly, compounds having a steroid-based structure (i.e., sterol lipids, sterols, steroids, androgens, estrogens and compounds in the cholesterol or ergosterol pathway) are significantly enriched in the “best E3” subsets of all three proteins, i.e., both the *Hs*σ1-R structures and the ERG2 model, with respect to categories comprising compounds with very diverse chemical structures, such as metabolites and compounds approved for clinical use by the FDA or other regulatory agencies. In detail, steroid-based compounds are less than 25% of the total number of compounds in the 21,359 compounds dataset used for VS, and 61%, 69% and 47% of compounds among the “best-E3” subsets of results against the *Hs*σ1-R structure in coordinate files 5HK1 and 6DK1 and the ERG2 molecular model, respectively, which corresponds to an enrichment in steroid-based compounds of 2.5, 2.8 and 1.9 folds, respectively.

Finally, all of the low-affinity *Hs*σ1R binders included in our dataset have an E_calc_ higher than that of all the compounds included in the “best-E3” subsets obtained from VS against *Hs*σ1-R structure in both coordinate files 5HK1 and 6DK1 (see Appendix A).

Taken together, these results indicate that the program used for VS has a good ability to recognize actual *Hs*σ1-R ligands, which are enriched among the “best-E3” hits of both the highest resolution coordinate file 5HK1 and the homologous ERG2 protein model (although not among the “best-E3” hits of the lower resolution coordinate file 6DK1), as well as to identify low-affinity *Hs*σ1-R binders, and that steroid-based compounds are among the *Hs*σ1-R preferred ligands.

### 2.3. Virtual Screening against Xlσ1-R Structures

To obtain further information about the nature of physiological binders of σ1-R proteins, we tried to identify the compound(s) giving rise to the electron density peak near the binding site of the *Xl*σ1-R structure in coordinate files 7W2B and 7W2E.

To this end, we first performed a VS of the 1332 yeast metabolites dataset (see Materials and Methods) against the two *Xl*σ1-R apo structures in coordinate files 7W2B_A and 7W2E_A. The results of this VS are summarized in Table 3; the full results for the “best-E3” subsets are reported in Appendix A, respectively, and plotted in Appendix A, respectively.

As reported for the “best-E3” subsets obtained from VS against *Hs*σ1-R structures in coordinate files 5HK1 and 6DK1, a lack of correlation between the E_calc_ of the 88 common “best-E3” hits is also observed for the results of VS against *Xl*σ1-R structures in coordinate files 7W2B and 7W2E (Appendix A).

To select compounds likely to fit in the electron density maps in the ligand binding site of *Xl*σ1-R structures, we visually inspected the 2D structures of the 88 “best-E3” hits that are common between the results of VS against the *Xl*σ1-R structures in coordinate files 7W2B and 7W2E. We selected five compounds (Table 4) based on the following criteria: (i) the compatibility of their molecular shape with the electron density peaks observed in the apo 7W2B and 7W2E structures; and (ii) the fact that their molecular structures were quite different from one another and, at the same time, each of them was similar to other compounds satisfying the first criteria.

### 2.4. Fitting of Selected Compounds into Xlσ1-R Electron Density

For both *Xl*σ1-R structures, we selected the chain where the electron density map peak found in the proximity of the *Xl*σ1-R binding site in the Fo-Fc map is most intense, namely chain C for both coordinate files 7W2B (7W2B_C) and 7W2E (7W2E_C). Then, we tested the selected compounds for their ability to fit the electron density peak in 7W2B_C and 7W2E_C, and calculated the average B-factor values of the resulting complexes (Table 4). Comparison of these values with those calculated for each chain of coordinate files 7W2E and 7W2B in the absence of ligands (Table 5) shows that they are in the same order of magnitude. Additionally, visual inspection of the generated complexes (Figure 4) indicated that the five selected compounds fit very well in the electron density map peak of both 7W2B_C and 7W2E_C. In line with the enrichment in steroid-based compounds in the “best-E3” results of VS experiments, ergosterol was the compound giving rise to the lowest B-factor value in the complex with coordinate file 7W2E_C and the second lowest B-factor value in complex with 7W2B_C. As shown in Figure 4 (panels B and H), the four A-D rings making the steroid nucleus are within the electron density peak, with only part of the ergosterol long chain substituent at position 17 falling outside the electron density.

### 2.5. In Vitro Assessment of Direct Ligand Binding to Hsσ1-R by Fluorescence Spectroscopy

Based on the results of VS experiments and electron density map fitting, indicating that compounds comprising a steroid nucleus are likely to be among the preferred σ1-R ligands, we inspected the results of VS against *Hs*σ1-R in coordinate sets 5HK1 and 6DK1 (Appendix A), to identify a steroid-based compound suitable for experimental assessment of *Hs*σ1-R binding ability. We selected 16,17-didehydroprogesterone (LIPID MAPS ID: LMST02030163), because: (i) it is the compound with the lowest E_calc_ among the “best-E3” hits of VS against coordinate file 5HK1 comprising a steroid nucleus; (ii) it is a human endogenous compound; and (iii) it has a very short chain substituent at position 17. The molecular model of the complex between the *Hs*σ1-R in coordinate file 5HK1 and 16,17-didehydroprogesterone is shown in Figure 5. Examination of the σ1-R residues at a distance ≤ 4.0 Å from the ligand reveals that the carbonyl oxygen in position 3 of 16,17-didehydroprogesterone may establish a polar interaction with the side-chain carboxylic group of E172, in the protonated state, thus replacing the basic amino group shared by classic pharmacophoric models. The non-polar remaining regions of 16,17-didehydroprogesterone establish hydrophobic interactions with hydrophobic residues lining the ligand binding site (i.e., V84, W89, M93, Y103, L105, F107, W164, I178, L182, A185 and Y206), most of which are the same residues that interact with ligands present in experimentally determined structures. Additionally, the carbonyl oxygen in position 20 of 16,17-didehydroprogesterone may establish a polar interaction with the side-chain hydroxylic group of T181.

To assess the ability of 16,17-didehydroprogesterone to bind *Hs*σ1-R, we took advantage of the presence of eleven tryptophan residues within the protein (Appendix A) to implement a fluorescence spectroscopy assay. We found that *Hs*σ1-R shows a strong intrinsic fluorescence at 340 nm when excited at 280 nm. Since the intrinsic fluorescence of tryptophan residues within a given protein depends on the tryptophan’s environment, when a molecule binds near tryptophan residues it can lead to fluorescence intensity quenching (Figure 6). Thanks to the fact that two of the tryptophan residues (i.e., Trp89 and Trp164) are part of the previously identified *Hs*σ1-R binding site (Appendix A), we were able to perform fluorescence titration to measure the affinity of selected molecules for this receptor. To validate the method, we performed fluorescence titration using pridopidine and iloperidone in addition to 16,17-didehydroprogesterone (Figure 6), since both molecules have been previously demonstrated to bind *Hs*σ1-R with high affinity using different techniques [11,27].

Pridopidine was initially shown, by 3H](+)-pentazocine displacement experiments [27], to have a K_i_ value for *Hs*σ1-R in the nanomolar range (81.7 nM). Subsequently, we reported that both pridopidine and iloperidone have K_D_ values towards *Hs*σ1-R in the micromolar range, as measured by surface plasmon resonance (SPR) experiments [11].

To determine K_D_ values (see Table 6), fluorescence titration data were fitted according to Equations (1) and (2) (see Section 4). Data analysis suggested that *Hs*σ1-R has two binding sites for the examined ligands, i.e., one high-affinity site and one low-affinity site. The K_D_ values of pridopidine and iloperidone for the *Hs*σ1-R high-affinity site are 254 and 19 nM, and therefore they are both higher than those previously measured by SPR (i.e., 15 and 5 μM, respectively), although in both cases iloperidone resulted to have a higher affinity for *Hs*σ1-R [11]. The K_D_ value of 16,17-didehydroprogesterone for the high affinity site was 10 nM, very similar to that of iloperidone and 25-fold better than that of pridopidine, indicating that 16,17-didehydroprogesterone is a very high affinity ligand for *Hs*σ1-R.

## 3. Discussion

The ER-resident σ1-R is being intensively studied because of its involvement in several CNS disorders and because of the neuroprotective activity of its agonists. Many experimental and computational studies have provided valuable information on putative entrance and exit pathways to and from the ligand binding site, and on a number of compounds able to bind σ1-R and elicit or inhibit specific activities. However, two essential receptor features, such as the route of ligand access to the binding site and the nature of the physiological ligand(s), have not yet been unequivocally determined. On the other hand, both pieces of information would be required to both understand the physio-pathological role of σ1-R and to design novel, higher affinity and higher specificity ligands endowed with specific agonistic or antagonistic activities.

To shed light on these two aspects, in this work we performed molecular dynamics simulations and investigated ligand binding to σ1-R by a combination of VS, electron density map fitting and fluorescence titrations.

To examine the pathway of ligand entry and exit from the ligand binding site, we performed MD simulations in different conditions with respect to those reported in previous studies (CIT Nb. 5,12). First, we chose the trimeric structure of *Hs*σ1-R, as opposed to the monomer used in previous simulations, because the trimer is the minimal quaternary assembly that is present in all *Hs*σ1-R and *Xl*σ1-R structures. Additionally, to avoid introducing any bias and only observe the behavior of the receptor, the simulation was performed without the application of external forces and in the absence of ligands placed at arbitrarily chosen positions outside of the protein. In this regard, one possible way to investigate the ligand entry pathway(s) is represented by the simulation of the ligand-receptor complex dissociation. However, this would necessarily require the use of constrained MD simulations, resulting in the introduction of bias in our simulation. A second type of bias would be introduced by simulating the dissociation of a specific ligand from the receptor, due to the dependence on the arbitrary choice of a specific ligand other than the physiological one (that is currently unknown), since the behavior of the chosen ligand might differ from that of the physiological ligand. Further, we chose conventional MD simulations rather than the aMD used in previous work [5]. The advantage of aMD consists of the fact that it allows protein conformational changes occurring on time scales inaccessible to conventional MD simulations to be investigated. However, as also pointed out before [28], while aMD gives us the possibility to enhance the exploration of conformational space, it does not reproduce the exact dynamics of the system. For this reason, conventional MD simulations should be preferred to aMD when they allow conformational changes occurring on an accessible time scale to be identified. The absence of any external force in the simulation presented in the present work excludes the fact that such conformational changes may be due to artifacts caused by the application of an artificial force, thus giving more credit to our findings.

In our system, while the overall secondary structures of the three monomers were substantially stable during the simulation time, significant conformational changes occurred in two regions of monomer B flanking the ligand binding site, and led to the opening of a cavity between the ligand binding site and the lipid bilayer (Figure 7). The first of these regions comprises residues 115–128, including part of the β4 and β5 strands and the loop comprised between them, and the second region comprises residues 172–188, including part of the β10 strand and the α4 helix. These results are in partial agreement with the results of previous MD simulations performed on *Hs*σ1-R, which highlighted structure alterations affecting residues E123 (β5) and R175 (β10) [5], each of which is comprised in one of the regions that unfolds in our studies. However, our results indicate that ligand entrance and exit occur via the protein side leaning on the membrane (pathway 2), whereas both previous MD studies point to an opening towards the aqueous medium (pathway 1) [5,12]. The hypothesis that the ligands enter and exit the binding site from the membrane side is in agreement with experimental studies on *Xl*σ1-R ligand binding site occupancy, in conditions where either the aqueous medium entrance (pathway 1) or the membrane (pathway 2) were alternatively blocked [13]. Additionally, a dynamic behavior of the α4 helix has been observed in the only *Hs*σ1-R structure determined in complex with a classic agonist molecule (i.e., (+)-pentazocine) [5] and in all *Xl*σ1-R structures determined in the open-like conformation [13]. Further, all known σ1-R ligands, agonists and antagonists alike, have a strong hydrophobic character and are therefore likely to have a high lipid/water partition coefficient. It is worth noting that previous aMD studies were performed on the *Hs*σ1-R in several conditions, including the absence of ligand, the presence of (+)-pentazocine or haloperidol in the position found in the crystal structures, and the presence of (+)-pentazocine placed at a distance > 10 Å from the binding site and inside the aqueous medium; however, they were not performed with the ligand placed at the same distance from the binding site but on the opposite side, inside the membrane. It would be interesting to investigate whether such a simulation would reveal an access route from the membrane side as well. 

It is worth reminding that only a single, 1500 ns-long, all-atoms MD simulation of the *Hs*σ1-R trimer embedded in the lipid bilayer was employed in this study. As is well known, several MD replicas should be employed to get convergence of structural, dynamical and energetic properties averaged over the phase space. However, this was not the aim of the present study. Here, we started from a structure far from the equilibrium (i.e., the crystal structure of the holo form of the *Hs*σ1-R trimer, from which we removed the ligand) to carry out a MD simulation in physiological-like conditions, long enough to identify conformational changes that might be associated with the opening of the ligand gate. Based on a single simulation, it is not possible to obtain information on features such as the kinetics of the binding pocket opening, or whether more than one monomer may open simultaneously. Nevertheless, our simulation was able to detect for the first time, without any constraint, in the trimer form embedded in a membrane reproducing the composition of ER, the conformational changes associated with the opening of the binding pocket for one of the three monomers. In our opinion, this result is of great significance for the understanding of the ligand binding mechanism in *Hs*σ1-R and the credibility of the previously suggested ligand entrance path [13]. More replicas would increase the sampling of the phase space, but they would not change the evidence reported in this work about the ligand entry pathway identified for one of the three monomers. Taken together, our results and those of previous studies support the hypothesis that the ligand enters and exits from the binding site through the membrane side of the protein (pathway 2). Nevertheless, while blocking experiments on *Xl*σ1-R demonstrate that ligands do access the binding site through pathway 2, they do not definitely exclude the fact that pathway 1 may be used as well [13]. As a consequence of this, of the results of previous MD simulations on *Hs*σ1-R, of the fact that a single MD simulation is reported in this work and additional MD trajectories could in principle identify other possible opening mechanisms of the binding site, and of the amphiphilic nature of the ligands, the possibility that ligands access the binding site through pathway 1 as well, possibly as a secondary route and/or in specific conditions, cannot be excluded.

To investigate the structure of *Hs*σ1-R physiological agonist(s), we applied both computational and experimental procedures. First, we computationally screened a large library of compounds, comprising human metabolites, against the experimental structure of *Hs*σ1-R determined either with highest resolution or in complex with an agonist molecule, to try and identify recurring structures among ligands likely to bind the receptor with high affinity. For comparison purposes, we screened the same library against the molecular model of the ERG2 protein, whose reaction substrate and product are known. VS results were sorted based on their predicted interaction energy (E_calc_) with *Hs*σ1-R. Since individual E_calc_ values are expected to have a 2–3 kcal/mol standard deviation from the actual protein–ligand interaction energy, we focused our analysis of the results of each VS on the set of hits whose E_calc_ value was within one standard deviation from that of the best hit (“best-E3” subsets). Examination of the categories of ligands included among the “best-E3” subsets resulting from VS against two *Hs*σ1-R structures and the ERG2 model highlighted a significant enrichment in compounds comprising a steroid scaffold with respect to the whole set of compounds used for VS (2.8 percentage increase towards the highest resolution *Hs*σ1-R structure), as opposed to those categories of compounds comprising molecules with very diverse structures, namely: (i) the large category comprising all human metabolites, which was significantly under-represented within the “best-E3” subsets resulting from VS against all three coordinate files (i.e., the two *Hs*σ1-R structures and the ERG2 model); and (ii) the compounds approved for clinical use by the FDA or other regulatory agencies, whose percentage was either slightly increased or slightly decreased within the “best-E3” hits resulting from VS, depending on the target structure. Conversely, the category of experimentally validated σ1-R ligands was shown to be significantly enriched among the “best-E3” hits of the VS against the highest resolution *Hs*σ1-R structure and the ERG2 model. Further, compounds experimentally shown to bind *Hs*σ1-R with very low affinity were not comprised in the “best-E3” subsets resulting from VS against *Hs*σ1-R in coordinate files 5HK1 or 6DK1. These results indicate that, in the VS performed in this work, the ligand binding sites of the *Hs*σ1-R highest resolution structure and of the ERG2 model have a clear preference for steroid-based structures. Next, to get additional clues about putative physiological σ1-R ligands, we took advantage of the two recently determined apo structures of *Xl*σ1-R that showed an electron density peak in the ligand binding site. We performed a VS against both structures using a library of ligands comprising all yeast metabolites with MW ≤ 400 Da, because the *Xl*σ1-R protein used for X-ray studies was expressed in and purified from yeast, and because inspection of the electron density map indicated that the yeast metabolite giving rise to the unfitted electron density was not larger than a cholesterol molecule. Visual inspection of the 88 compounds that resulted in the “best-E3” subsets obtained from VS against both structures led us to select five structurally diverse molecules that were likely to best fit the experimentally determined electron density on the basis of both their size and shape. According to both visual inspection and measurement of average B-factor values, and in agreement with these selection criteria, all five molecules were shown to fit well in the electron density map, within the ligand binding site of both the closed and the open-like form of apo *Xl*σ1-R, ergosterol being the best fitting compound.

Since both VS against *Hs*σ1-R structures and fitting into the electron density map of *Xl*σ1-R structures indicated steroid-based molecules as preferred σ1-R ligands, we decided to measure experimentally the affinity of one steroid-based compound against human *Hs*σ1-R. We selected 16,17-didehydroprogesterone because it is a human endogenous compound, it does not contain long substituents that may affect σ1-R binding and it is within the group of hits predicted by VS procedures to bind *Hs*σ1-R with highest affinity. To this end, we implemented a fluorescence titration procedure and used both pridopidine and iloperidone as positive controls. This test presents several advantages with respect to previous methods used to measure ligand affinity to σ1-R. Like surface plasmon resonance (SPR) experiments, it evaluates the interaction between a ligand and the purified receptor, therefore it is more direct than classic radioactive ligand displacement assays performed on membrane extracts, and does not present the disadvantages of the latter, such as the dependency of the measured affinity constant values on the specific radioactive ligand used in the experiment, and the possibility that even the most prototypical radioactive ligands to be displaced and/or the new ligands undergoing investigation bind to several receptors, in addition to σ1-R. Additionally, it is more realistic than SPR methods, since in the latter the σ1-R has to be immobilized on a solid matrix.

Fluorescence data measurements are compatible with the existence of two independent *Hs*σ1-R binding sites for the examined ligands, one with a high-affinity site and one with low affinity. The simplest explanation for this behavior is that both of the binding sites are present within the protein monomer. However, it cannot be excluded that a more complex mechanism of binding is at play, involving different oligomeric states that, as reported before [4], are not only heterogeneous but also variable upon ligand binding. For comparison with previously reported data, we discuss the K_D_ values obtained for the high-affinity site (K_D1_ in Table 6) only.

The K_D1_ values measured by fluorescence titration for pridopidine was around 250 nM, which is in between the K_D_ value of 81.7 nM determined in a [3H](+)-pentazocine displacement assay in cell membranes [27], and that of 15 µM, determined by SPR experiments, where ligand and purified σ1R immobilized on a matrix were allowed to interact directly [11]. In the fluorescence titration assay, the K_D1_ value of iloperidone was 19 nM, more than one order of magnitude lower than that of pridopidine, whereas in previous SPR experiments it had a K_D_ value of 5 µM, one third of that of pridopidine in the same assay; this indicates that, even if the absolute values are different, in both assays iloperidone is a better *Hs*σ1-R binder than pridopidine. According to fluorescence titration values, 16,17-didehydroprogesterone is an even better *Hs*σ1-R ligand than iloperidone and pridopidine, the K_D1_ value for the high-affinity site being 10 nM, which is two orders of magnitude lower than that of pridopidine for the same site in the same assay. The K_D1_ value reported here for the interaction between 16,17-didehydroprogesterone and *Hs*σ1-R is also one order of magnitude lower than that measured for progesterone by ligand displacement assays on guinea pig and rat brains, where progesterone was reported to be the steroid-based compound with the highest σ1-R affinity among those tested [29,30,31].

Analysis of the complex between *Hs*σ1-R and 16,17-didehydroprogesterone built by the VINA program indicates that the interaction mode of the ligand with the receptor was very similar to that observed for the ligands present in experimentally determined structures of complexes with *Hs*σ1-R, and to the shared features of pharmacophoric models. The main difference is in the replacement of the basic amino group shared by those ligands and pharmacophoric models with the carbonyl oxygen at position 3 of 16,17-didehydroprogesterone in the polar interaction with the conserved E172. In this model, the carbonyl oxygen of the ligand is expected to act as the electron donor and the side-chain carboxylic group of E172 is expected to be protonated and act as an electron acceptor. Given the results of this work and the well-known ability of steroid-based molecules to act as σ1-R agonists or antagonists, we suggest that pharmacophoric models for *Hs*σ1-R ligands should be expanded to include an oxygen-atom-containing group, with the aim to establish a polar interaction with E172, as an alternative to a basic nitrogen.

It is worth remarking that, in spite of the high overall similarity between the two *Hs*σ1-R structures (the RMSD values calculated after optimal superposition of residues in the C-term ligand-binding domain are in the range 0.4–0.6 Å for Cα atoms and 0.6–0.9 Å for all atoms) and between the two apo *Xl*σ1-R structures (Appendix A), there was no correlation between the E_calc_ values between the “best-E3” hits of 5HK1 and 6DK1 or those of 7W2B and 7W2E. Although it is possible that a correlation would be found if the ligand poses resulting from VS were subjected to energy minimization followed by VINA local search re-scoring, the results presented in this work suggest that the choice of the target structure, when more than one is available, affects the results of VS. Therefore, the “refinement” of experimental structures using methods such as energy minimization and molecular dynamics simulations as a propaedeutic step to VS, would better be avoided, at least until these methods become able to routinely provide conformations closer to the real structures than those determined with experimental methods.

## 4. Materials and Methods

### 4.1. Structure Selection, Visualization and Analysis

The 3D structures of human and *X. laevis* σ1-R used in this work were downloaded from the Protein Data Bank (PDB: https://www.rcsb.org/; accessed on 18 March 2022) [8].

Among the *Hs*σ1-R structures available from the PDB (see Table 1 in [11]*,* the structure in coordinate files 5HK1, determined in complex with the PD144418 antagonist, which has been solved with the highest resolution (2.51 Å) [5], was used for both molecular dynamics simulations and virtual screening (VS) experiments. Additionally, the 3D structure in coordinate files 6DK1, which is the only available Hsσ1-R structure that has been determined in complex with a classic agonist, namely (+)-pentazocine [5]), was also used for VS experiments, to investigate whether VS results are affected by small side-chain variations occurring upon binding with compounds endowed with different biological activities.

Among the *Xl*σ1-R structures available from the PDB (see Appendix A), we used the 3D structures in coordinate files 7W2E and 7W2B, because they both display an electron density map peak in the ligand binding site, due to a not yet identified ligand [13]. Interestingly, while the 7W1B structure was determined in a “closed” conformation, like that observed in *Hs*σ1-R structures, the 7W1E structure displayed an “open-like” conformation, where the α4 helix rotates slightly away from α5, thus enlarging an opening that may allow ligand entry.

For each of these structures, we selected chain A, after verifying that the structure of the ligand binding domain was conserved in all of the structure determination of the same protein. In the case of the A chain of coordinate files 5HK1 and 6DK1, we have previously reported that the conformation of the ligand binding domain (comprising residues 29–212 in 5HK2 chain B and 35–218 in all other chains) is highly conserved among the 15 chains included in the five structures. In the case of the A chain in coordinate files 7W2E and 7W2B, we compared all of the monomers present in all of *Xl*σ1-R structures to one another, and found that the ligand binding domain (comprising residues 31–216 in all monomers) is also highly conserved among the 50 monomers included in the seven structures (see Appendix A). For all monomer pairs, the RMSD values measured after optimal structure superpositions were ≤0.6 Å for Cα atom and ≤1.05 Å for all atoms, the highest values resulting from comparison of monomers with an “open-like” conformation with monomers with a “closed” conformation. The RMSD values calculated following pair-wise superposition of “open-like” monomers with one another were ≤0.25 Å for Cα atoms and ≤0.42 Å for all atoms. The RMSD values calculated following pair-wise superposition of “closed” wild-type monomers with one another were ≤0.14 Å for Cα atoms and ≤0.28 Å for all atoms, and 0.36 Å for Cα atoms and 0.74 Å for all atoms in case the double-cysteine mutant in coordinate file 7W2H was included in the comparison.

Structure visualization and analysis were performed using the programs InsightII [32], Swiss-PDBViewer [33], CHIMERA [34] and PyMol [35].

### 4.2. Molecular Modelling

The atomic model of the ERG2 protein from yeast was downloaded from the AlphaFold protein database [36,37].

### 4.3. Molecular Dynamics

The coordinates of the bound PD144418 *Hs*σ1-R antagonist were removed from the 3D structure of *Hs*σ1-R in coordinate file 5HK1, to generate the apo form [1] The AMBER03 force field [38] was used to describe bonded and non-bonded interactions of protein residues. The apo form of *Hs*σ1-R was embedded into a membrane bilayer consisting of 466 POPC, 147 POPE, 83 POPG and 54 cholesterol lipids, thus reproducing with good approximation the composition of the ER membrane [39].

The membrane bilayer was generated using the webserver MemGen (http://memgen.uni-goettingen.de; accessed on 1 February 2021) [40]. The system, containing 466 POPC, 147 POPE, 83 POPG and 54 cholesterol lipids, 67,210 water molecules, and Na^+^ and Cl^−^ ions at a concentration of 0.15 M, with an excess of Na+ ions to compensate for the net negative charge of the lipids, for a total of about 304,000 atoms, was equilibrated for 200 ns by means of all-atoms classical MD simulations, with a time step of 2 fs for numerical integration. The temperature was kept constant at 298 K using the Nose–Hoover algorithm [41,42] with a coupling time constant τT = 0.4 ps. The system was semi-isotropically coupled to a pressure bath at 1 bar with τP = 1.0 ps, using the Berendsen barostat [43]. The lipids were described with Slipids/AMBER (FF) Parameters [44]. The last frame of the MD simulation was used to embed the protein in the lipid bilayer using the “membed” module as implemented in the “mdrun” program of the Gromacs package 2018.3, with a resize factor of 0.7 in the xy plane.

The protein embedded in the lipid bilayer was solvated in a box with dimensions 15.2 × 15.2 × 13.6 nm (see Figure 1), using the TIP3p water model [45]. About 71,000 water molecules were added to the protein/membrane system. Na^+^ and Cl^−^ ions were added to the solution at a concentration of 0.15 M with excess Na^+^ ions to compensate for the net negative charge of the protein/membrane system. The protonation states of titratable residues present in the protein were assigned based on empirical pKa prediction using the PropKa program [46]. The proton position of neutral histidine residues was chosen by visual inspection of the structure. Finally, the simulated system was composed of almost 320,000 atoms. MD simulations were carried out using the GROMACS 3.0 software package [47,48]. Long-range electrostatic interactions were computed by the Particle Mesh Ewald (PME) method [49], using a grid spacing of 0.12 nm and a short-range cut-off of 1.2 nm. The LINCS algorithm [50] was applied to constrain the bond lengths of the hydrogen atoms to a constant value. A time step of 2 fs was used for numerical integration of the equations of motion. The temperature was kept constant at 310 K using the V-rescale algorithm with a coupling time constant τ_T_ = 0.1 ps [51] and the system was semi-isotropically coupled to a pressure bath at 1 bar with τ_P_ = 1.0 ps, using the Parrinello–Rahman barostat [52,53].

A 180 ns long MD simulation with harmonic position restraints (force constant 1000 kJ mol^−1^ nm^−2^) on the heavy atoms of the protein residues was carried out before the unrestrained MD simulation, using the same setup described above. The only difference was the replacement of the Parrinello–Rahman barostat with the Berendsen barostat in the position restrained simulation [43]. Finally, 1.5 µs of unrestrained MD simulation was carried out.

### 4.4. Virtual Screening

#### 4.4.1. Receptor Preparation

Crystallographic waters and ligand molecules were removed using Chimera. AutoDock Tools (ADT) v. 1.5.6 was used to add hydrogen atoms, merge non-polar hydrogen atoms and automatically assign Gasteiger charges.

#### 4.4.2. Choice of Ligand Datasets

For virtual screening against the *Hs*σ1-R structure in coordinate files 5HK1_A or 6DK1_A or against yeast ERG2 model, we combined the datasets listed below in a single comprehensive dataset.

From the ZINC15 database (http://zinc15.docking.org/; accessed on 1 June 2020) [25] we selected the large set of molecules tagged as “metabolites” (15,871 compounds), which in principle should include the physiological ligand of *Hs*σ1-R. Additionally, from the same database we selected all molecules tagged as “FDA approved” or “World not-FDA” (1538 and 3192 compounds, respectively), which have been approved as drugs by the FDA or other regulatory agencies. The latter set of compounds was included because compounds approved by regulatory agencies worldwide and predicted to bind σ1-R with high affinity by VS procedures may be considered as candidates for “repositioning” as neuroprotective agents; moreover, this set included six FDA-approved drugs that we have recently demonstrated to be able to directly bind purified *Hs*σ1-R in vitro and improve the growth of HD cells from both or one HD patient [11], which can act as positive controls. For all these sets, the option “for sale” was also selected, to ensure that compounds of particular interest could be experimentally tested.

Due to the demonstrated ability of neurosteroids to act as σ1-R agonists or antagonists [23], and the fact that the homologous ERG2 protein present in fungi binds steroid-based compounds, we selected all the compounds belonging to the sterol_lipids, sterols, steroids, androgens, estrogens, cholesterol or ergosterol biosynthetic pathway categories and available for sale from the LIPID MAPS database (https://www.lipidmaps.org/; accessed on 1 July 2021) [26], comprising 3761, 1593, 362, 101, 63, 442 and 347 compounds, respectively.

Finally, from literature searches and the available 3D structures of human and *X. laevis* σ1-R, we obtained a list of 36 experimentally validated σ1-R and/or σ2-R binders, to use as positive controls. σ2-R is a receptor located in the ER and involved in diseases including neurological diseases and cancer [54]. In spite of the fact that the two receptors have different folds (σ2-R is a four-pass transmembrane protein), σ1-R and σ2-R have been shown to share several ligands [55,56].

The final comprehensive dataset comprised 21,359 non-redundant compounds (several compounds are present in more than one of the listed categories). The structures of these compounds were downloaded from ZINC, whenever available, or from the ChEbi [57] or PubChem [58] databases.

For VS against *Xl*σ1-R A chain monomers in coordinate files 7W2E or 7W2B (i.e., 7W2E_A and 7W2B_A, respectively), or against the yeast ERG2 model, we used a 1332 molecule dataset obtained from the yeast metabolome database (YMDB) [59] by eliminating all compounds whose molecular weight was higher than 400 Da. This was done to speed up the VS procedure, since a preliminary visual inspection of the density map in the ligand binding site of 7W2E_A and 7W2B_A had revealed that the unknown *Xl*σ1-R ligand in these two apo structures was not larger than a cholesterol molecule, which has a molecular weight of about 387 Da.

#### 4.4.3. Ligand Preparation

All compounds present in the selected datasets were converted: (i) from the .smi or .sdf to the PDB format, using an ad hoc developed bash script that included the “molconvert” command from MarvinSketch v18.26 (https://chemaxon.com, accessed on 1 July 2021); (ii) from the PDB to the pdbqt format, using a script from AutoDock Tools v1.5.6 (ADT) [60] where the following parameters were added: “-A ‘hydrogen_bonds’” to both add hydrogens and build bonds among non-bonded atoms; and “-U ‘nphs_lps’” to merge both non-polar hydrogens and lone pairs.

### 4.5. Virtual Screening

For VS against *Hs*σ1-R structures in coordinate files 5HK1_A and 6DK1_A, the following space searching parameters were adopted: spacing value at 0.375 Å; center on coordinates 12.168, 36.423 and −34.778; and 30 × 24 × 34 grid points. 

For virtual screening against *Xl*σ1-R monomers 7W2E_A and 7W2B_A, only coordinates and dimension of the binding pocket were changed: center on coordinates −31.000, −26.000 and 34.000; and 12 × 12 × 12 grid points.

VS was performed using the program VINA [61] and the same parameters for all four σ1-R structures and the ERG2 model, namely: “--num_modes 100”, which represents the maximum number of binding modes to generate; and “--energy_range 9”, in order to maximize the energy difference between the best binding mode and the worst one. Additionally, all conformations (poses) were kept, rather than only those with a VINA score better than a given threshold. All other parameters had default values. After VS, the “vina_screen_get_top.py“ script from AutoDock Vina tools [61] was used to extract the receptor–ligand interaction energy (E_calc_) from the pdbqt files created by VINA, and sort hits on the bases of E_calc_ values. 

Previously developed Python scripts were used to parse VINA output files and perform a preliminary analysis of the selected ligand–receptor complexes. In particular, the pose energy of each ligand was extracted from the VINA pdbqt file; features of protein–ligand interactions such as hydrogen bonds, number of contacts and number of unfavorable interactions (clashes) were calculated by the structure visualization and analysis program Chimera, following re-building of receptor–ligand complexes. Information on the clinical indication and mechanism of action of each compound was manually obtained from KEGG [62] and DrugBank [63].

### 4.6. Fitting of Compounds into Xlσ1-R Electron Density

The coordinates and the structure factor files of the *Xl*σ1-R structure in the apo forms, namely 7W2B (“closed” conformation) and 7W2E (“open-like” conformation), were downloaded from the Protein Data Bank (https://www.rcsb.org/; accessed on 1 December 2021). Standard procedures to refine structures of complexes between proteins and small molecules were employed to perform the fitting of selected compounds within the unassigned electron density in the ligand binding site of *Xl*σ1R structures. The program COOT [64] was used to fit selected compounds in the Fo-Fc electron density map present in the binding site of both structures, and the program REFMAC [65] of the CCP4i program suite [66,67] was used to refine the resulting complex structures. The crystallographic information file (.cif) and the coordinate files for each selected compound (.pdb) were obtained starting from the corresponding smiles found in the database using the program ACEDRG [68].

### 4.7. Hsσ1-R Expression and Purification

The *Hs*σ1-R cloned into a pFastbac1 plasmid encoding for the FLAG tagged *Hs*σ1-R gene was kindly provided by Prof. Andrew Kruse (Harvard Medical School, Boston, MA, USA). The receptor was expressed according to previously described conditions at the Protein Facility of Elettra Sincrotrone Trieste (Basovizza, Trieste, Italy) [11]. The bacmid was generated by transposition using *E. coli* DH10Bac competent cells (Invitrogen, Waltham, MA, USA). Sf9 insect cells (Expression Systems, Davis, CA, USA) were used for virus preparation and protein expression. Infection was performed at a density of 1 × 10^6^ cells/mL and cultures were grown at 27 °C for 72 h. Cells were harvested by centrifugation and frozen at −80 °C until purification. The protein was purified according with a protocol developed by Schmidt et al. [1].

After thawing frozen cell paste, cells were lysed by osmotic shock in 20 mM HEPES pH 7.5, 2 mM magnesium chloride and 1:100.000 (*v*:*v*) benzonase nuclease (Merck, Darmstadt, Germany), and stirred for 15′ at room temperature. Lysed cells were centrifuged for 1 h at 4 °C at 38,000× *g*. The protein in the pellet was extracted using a glass dounce tissue grinder in a solubilization buffer containing 250 mM NaCl, 20 mM HEPES pH 7.5, 20% glycerol, 1% *w*/*v* lauryl maltose neopentyl glycol and 0.1% *w*/*v* cholesteryl hemisuccinate (from premade solution LMNG/CHS 5%/0.5%, Anatrace). The sample was stirred for 2 h at 4 °C and subsequently centrifuged as described before. The filtered supernatant, containing solubilized receptor, was then loaded at a flow rate of 1 mL/min on anti-DYKDDDK resin (#L0043, GenScript, Nanjing, China), previously washed according to the manufacturer’s instructions and equilibrated with 20 mM Hepes pH 7.5, 100 mM NaCl, 0.2% *v*/*v* glycerol, 0.1% *w*/*v* LMNG and 0.01% *w*/*v* CHS (buffer A). The protein-loaded resin was first washed extensively with buffer A and then washed with buffer B (20 mM Hepes pH 7.5, 100 mM NaCl, 0.02% *v*/*v* glycerol, 0.01% *w*/*v* LMNG and 0.001% *w*/*v* CHS). Then, the protein was eluted with buffer B supplemented with 0.2 mg/mL Flag peptide. *Hs*σ1-R purity was assessed by SDS–Page. The receptor was further purified using size exclusion chromatography (SEC) on a Sephadex S200 column (GE Healthcare) in a buffer containing 0.01% LMNG, 0.001% CHS, 100 mM NaCl and 20 mM HEPES pH 7.5. The elution peak corresponded to a decamer (see Appendix A).

After this last purification step, the receptor was flash frozen in liquid nitrogen and stored at −80 °C until use for experiments.

### 4.8. In Vitro Assessment of Direct Ligand Binding to Hsσ1-R by Fluorescence Spectroscopy

Binding of ligands to *Hs*σ1-R was followed by measuring the intrinsic protein fluorescence on a Fluoromax-4 spectrofluorometer (Horiba-Jobin Yvon). Fluorescence spectra were collected at 25 °C using a 1 cm path length cuvette, under continuous stirring. The excitation wavelength was 280 nm, and emission was recorded between 300 and 450 nm. The protein was used at 100 nM concentration in a buffer containing 0.01% LMNG, 0.001% CHS, 100 mM NaCl and 20 mM HEPES pH 7.5. The ligands were added to the protein solution, starting from a 10 mM stock solution in 100% DMSO, at a final concentration within the range 0.01 to 100 μM. The measurements were performed after each addition, following 3 min of incubation. Among the tested ligands, i.e., 16,17-didehydroprogesterone, pridopidine and iloperidone, the latter was the only one that gave rise to an inner filter effect. This effect occurs if the chemical species present in a protein solution absorb light at excitation or emission wavelengths, leading to reduced fluorescence emission intensities. Iloperidone absorbs light at 280 nm, which is also the excitation wavelength. Therefore, the observed fluorescence data for iloperidone were corrected for the inner filter effect, using the following equation [69,70]:(1)$$F_{corr}=F_{obs}\times 10^{(A_{ex}+A_{em})/2}$$
where *F_corr_* and *F_obs_* are the corrected and measured fluorescence intensities, respectively, and *A_ex_* and *A_em_* represent the differences in the absorbance values of the sample upon the addition of the ligand at the excitation (280 nm) and emission (300−400 nm) wavelengths, respectively.

The dissociation constant values were obtained by fitting the experimental data using Equation (2), which is the sum of two hyperbolas describing saturation binding to two independent binding sites, with the program KaleidaGraph (KaleidaGraph, Version 4.5.4 for Windows. Synergy Software v5, Reading, PA, USA. www.synergy.com, accessed on 1 July 2021). This equation describes a two-site model in which the binding sites are independent of each other, and at each site the interaction is non-cooperative [71]:(2)$$\Delta F=\frac{\left({\Delta F}_{max1}\times {\left[L\right]}_{tot}\right)}{\left(K_{D1}+{\left[L\right]}_{tot}\right)}+\frac{\left({\Delta F}_{max2}\times {\left[L\right]}_{tot}\right)}{\left(K_{D2}+{\left[L\right]}_{tot}\right)}$$
where: Δ*F* is the fluorescence quenching due to ligand binding; Δ*F_max_* is the maximum fluorescence quenching possible in the experiment; [*L*]*_tot_* is the total concentration of the ligand; and *K_D1_* and *K_D2_* are the dissociation constants for the two sites.

## 5. Conclusions

The results reported herein indicate, on the one hand, that *Hs*σ1-R structures have a clear preference for steroid-based structures, based on VS calculations, and, on the other, that 16,17-didehydroprogesterone, being an endogenous human compound, may be a physiological *Hs*σ1-R agonist, in addition to the other previously tested steroids and neurosteroids. Additionally, we show that extensive VS experiments, combined with evolutionary analyses and electron density map fitting, may be useful tools to guide selection of potential protein ligands, especially in cases when actual ligands cannot be identified based on E_calc_ values provided by VS programs alone. Further, we present a plausible model of the interaction of *Hs*σ1-R with 16,17-didehydroprogesterone and, possibly, other steroid-based compounds, which are not included in previous pharmacophoric models. Finally, our MD simulations support the hypothesis that ligands preferentially access the σ1-R binding site from the membrane side of the protein (pathway 2).

The hypotheses that steroids are among the preferred σ1-R natural ligands, and that the main route of access of natural ligands to the σ1-R ligand binding site is from the membrane side of the protein, are both in agreement with and strengthen each other.

## Figures and Tables

**Figure 1 ijms-24-06367-f001:**
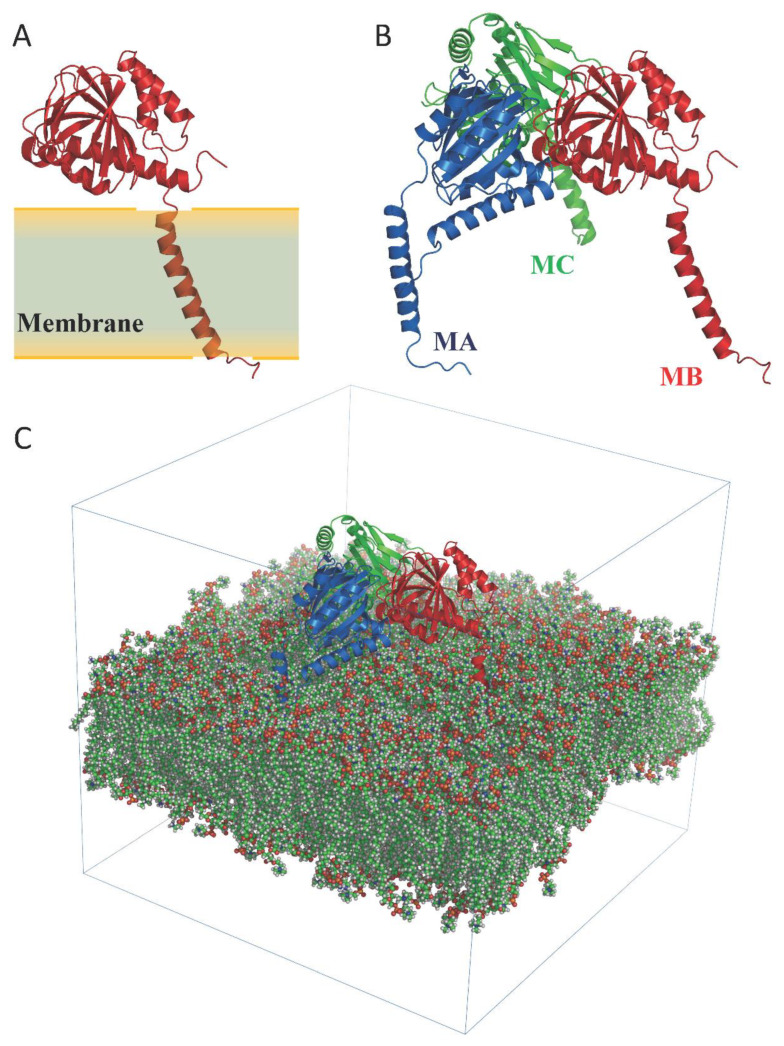
Structure of *Hs*σ1-R protein. (**A**) A single monomer is shown as ribbon. The position of the monomer with respect to the membrane region of the protein is highlighted. (**B**) The homotrimer is shown as ribbon. The three monomers (MA, MB and MC) are colored blue, red and green, respectively. (**C**) Simulated system within the simulation box. Membrane atoms are shown as spheres and colored by atom type: C, O, N, P and H are green, red, blue, orange and white, respectively.

**Figure 2 ijms-24-06367-f002:**
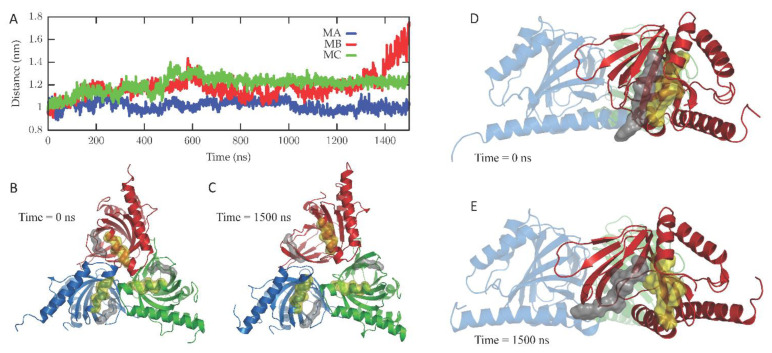
Conformational changes occurring in the *Hs*σ1-R protein along the simulated trajectory. (**A**) The distances between the mass centers of the backbone atoms of residues 118–121 and 180–188 for the three monomers (namely, MA, MB and MC) are reported as a function of time. (**B**,**C**) Cartoon representation of the *Hs*σ1-R protein viewed from the membrane side at 0 ns (**B**) and 1500 ns (**C**) of the MD simulation. The three monomers are colored blue (M1), red (M2) and green (M3). The surface of residues 118–121 and 180–188 of the three monomers is also shown and colored grey and yellow, respectively. (**D**,**E**) Cartoon representation of the *Hs*σ1-R protein C-terminal domains, external to the membrane, at 0 ns (**D**) and 1500 ns (**E**) of the MD simulation. The surface of residues 122–126 and 171–176 of M2 (red) is also shown and colored grey and yellow, respectively, highlighting the opening of the *Hs*σ1-R binding site that occurs along the simulated trajectory.

**Figure 3 ijms-24-06367-f003:**
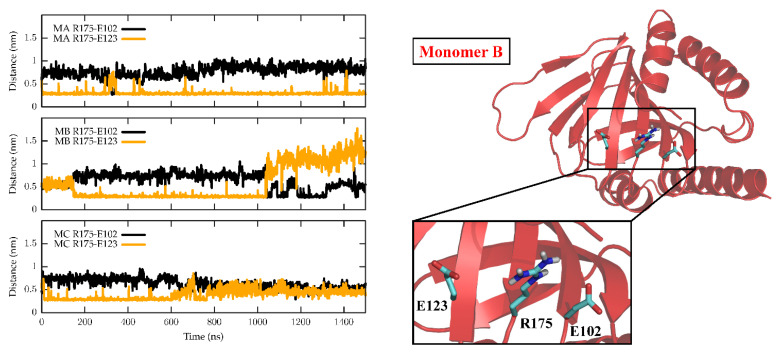
Time evolution of salt bridges between residues R175, E102 and E123 along the MD simulation. In the **left** panel, the minimum distances calculated between R175 and E102 (black) and between R175 and E123 (orange) are reported as a function of the simulated time for the three monomers, namely M1 (**top** panel), M2 (**middle** panel) and M3 (**bottom** panel). **Right** panel: cartoon representation of M2 in the starting conformation of the MD simulation, which is virtually identical to the crystallographic conformation. Zoomed-in inset: residues R175, E102 and E123 are shown as sticks and coloured by atom type: C, cyan; N, blue; O, red; H, white.

**Figure 4 ijms-24-06367-f004:**
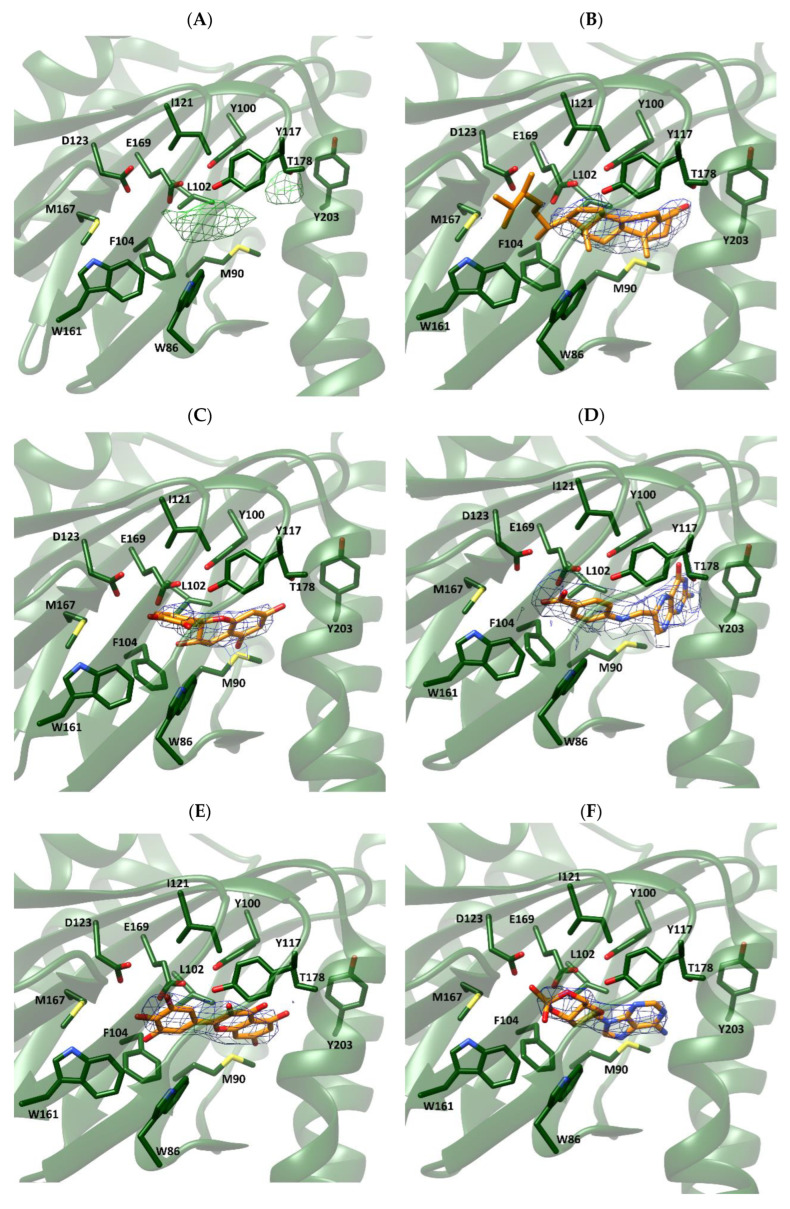
Fitting of selected compounds into the electron density within the ligand binding site of *Xl*σ1-R. The protein is shown as ribbon and colored green. The side-chains of residues surrounding the ligand binding site are shown as sticks and colored by atom-type: C, N, O and S atoms are green, blue, red and yellow, respectively. The structures in coordinate files 7W2B and 7W2E are shown in panels (**A**–**F**) and (**G**–**L**), respectively. Ligands in panels (**B**–**F**) and (**H**–**L**) are shown as sticks and colored by atom-type in the same way as protein side-chains, except that C atoms are orange. Ligands are: ergosterol, panels (**B**,**H**); catechin, panels (**C**,**I**); 7,8-dihydropteroic acid, panels (**D**,**J**); myricetin, panels (**E**,**K**); and 3′,5′-cyclic dAMP, panels (**F**,**L**).

**Figure 5 ijms-24-06367-f005:**
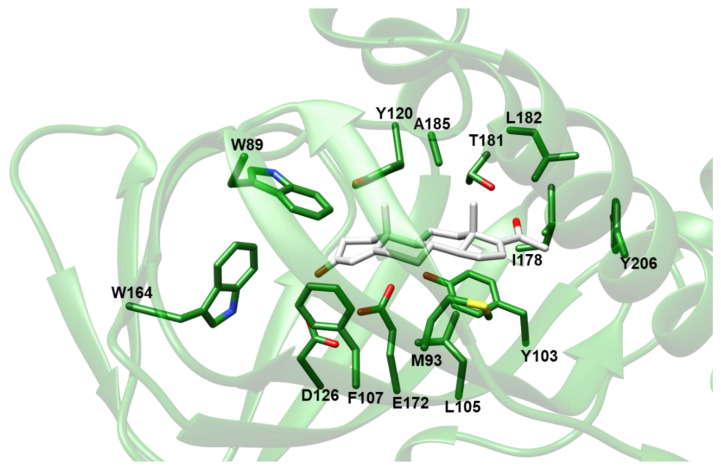
Molecular model of the complex between *Hs*σ1-R and 16,17-didehydroprogesterone built by VINA. The protein is shown as ribbon and colored green. The ligand and the side-chains of residues at a distance ≤ 4.0 Å from the ligand are shown as sticks and colored by atom-type: N, O and S atoms are blue, red and yellow, respectively; C is green for the protein and white for the ligand. The only exception is V84, which was removed from the picture for clarity.

**Figure 6 ijms-24-06367-f006:**
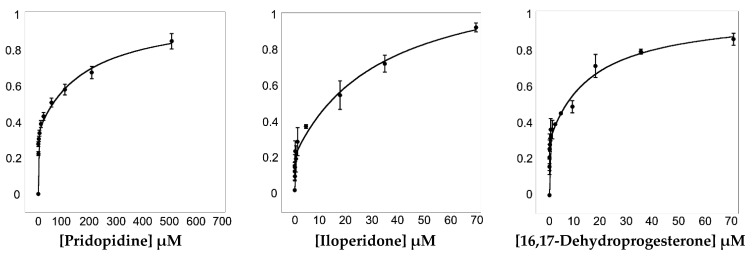
Titration of *Hs*σ1-R with pridopidine (**left** panel), iloperidone (**central** panel) and 16,17-didehydroprogesterone (**right** panel). ΔF/ΔF_max_ values at 340 nm are plotted as a function of ligand concentration (see Section 4). Fluorescence titration analyses were performed using Equation (2) that describes a two-site model in which both sites are independent of each other and non-cooperative. Data are from three replicate experiments. Error bars are standard deviations (SD) with n = 3.

**Figure 7 ijms-24-06367-f007:**
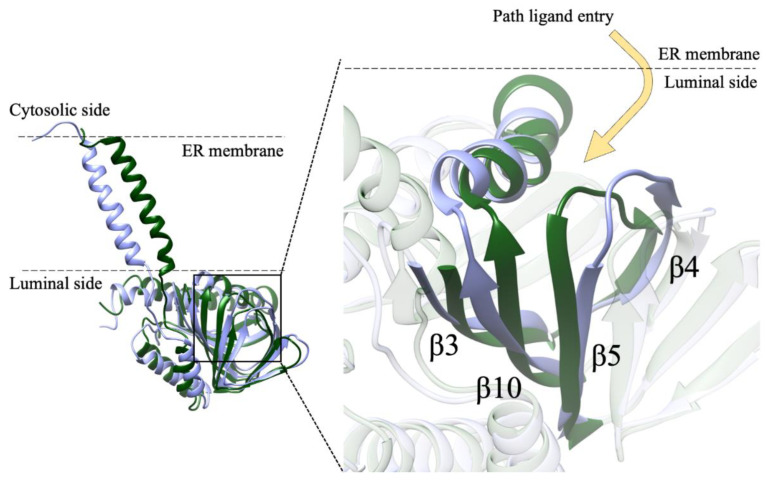
Path of ligand access to σ1-R binding site identified by MD simulations. Monomer B of *Hs*σ1-R in coordinate file 5HK1 at the beginning (t = 0 ns) and at the end (t = 1500 ns) of the MD simulation is shown as ribbon and colored green and lilac, respectively. Left: orientation of the monomer with respect to the ER membrane. Right: zoom-in of the image on the left. The ligand is proposed to access σ1-R from the protein side in contact with the ER membrane, either from the luminal medium or from the membrane itself. The movement of the α4 helix and of the β3, β4-β6 and β10 strands, comprising residues 102–109, 117–137 and 168–193, respectively, is clearly visible.

**Table 1 ijms-24-06367-t001:** Summary of the results of VS experiments against the 3D structures of *Hs*σ1-R in coordinate files 5HK1 and 6DK1 and the molecular model of yeast ERG2. 5HK1_A and 6DK1_A: monomer with chain ID “A” in coordinate files 5HK1 and 6DK1, respectively. ERG2: molecular model of yeast ERG2 protein built by AlphaFold2. B_20: 20 hits with lowest calculated interaction energy (E_calc_) with the target structure. E_3.0: “best-E3”, namely hits whose E_calc_ with the target structure is ≤3.0 kcal/mol higher than that of the best hit. Nb. Hits: number of hits in each results subset (i.e., B_20 and E_3.0). E_calc_ (kcal/mol): range of interaction energy with the target protein in each results subset. Hb, Contacts and Clashes: range of hydrogen bonds, overall contacts and unfavorable van der Waals contacts between ligand and target protein in each subject. Ago-Ant: known σ1-R and/or σ2-R binders. Metab, FDA and World: molecules tagged as “metabolites + for sale”, “FDA approved + for sale” and “World-not FDA + for sale” in the ZINC15 database [25]. Ste_Lip, Sterols, Steroids, Androg, Estrog, Chol_P and Ergo_P: molecules belonging to the sterol_lipids, sterols, steroids, androgens, estrogens, cholesterol or ergosterol biosynthetic pathway categories, respectively, and available for sale in the LIPID MAPS database [26].

	5HK1_A		6DK1_A		ERG2	
**Subset**	B_20	E_3.0	B_20	E_3.0	B_20	E_3.0
**Nb. Hits**	20	1679	20	3005	20	2574
**E_calc_ (kcal/mol)**	−13.1–(−12.3)	−13.1–(−10.1)	−13.2–(−12.5)	−13.2–(−10.5)	−11.7–(−11.0)	−11.7–(−8.7)
**Hb**	0–2	0–3	0–1	0–6	0–2	0–10
**Cont**	40–100	29–152	41–100	28–175	44–95	22–143
**Clashes**	0–4	0–7	0–3	0–7	0–3	0–8
**Ago-Ant**	0	8	1	5	0	10
**Metab**	2	431	3	681	10	1320
**Ste_Lip**	0	436	4	1192	4	627
**Sterols**	7	412	11	706	2	299
**Steroids**	0	109	0	138	1	218
**Androg**	0	52	0	68	1	79
**Estrog**	0	42	0	6	0	45
**Chol_P**	0	135	4	254	1	140
**Ergo_P**	1	124	1	169	0	85
**FDA**	3	140	0	144	0	312
**World**	8	248	1	261	4	43

**Table 2 ijms-24-06367-t002:** Comparison between the percentage of compounds in each category used for VS against the 3D structures of *Hs*σ1-R in coordinate files 5HK1 and 6DK1, or the molecular model of yeast ERG2, and the percentage of compounds in the same categories found in the “best-E3” subset obtained from VS against each structure. 5HK1_A and 6DK1_A: monomer with chain ID “A” in coordinate files 5HK1 and 6DK1, respectively. ERG2: molecular model of yeast ERG2 protein built by AlphaFold2. Category: category of compounds included in the complete 21,359 dataset used for VS against these structures. Ago-Ant, Metab, Ste_Lip, Sterols, Steroids, Androg, Estrog, Chol_P and Ergo_P, FDA and World are as in Table 1. All Ste, All Div and All Cat: all steroid-based compounds (comprising Ste_Lip, Sterols, Steroids, Androg, Estrog, Chol_P and Ergo_P), all compounds with diverse scaffolds (comprising Metab, FDA and World) and all compounds in all categories. Note that the sum of compounds in All Cat is higher than 21,359, which is the total number of compounds used for VS against 5HK1_A, 6DK1_A and ERG2, because many compounds belong to more than one category and, therefore, are counted more than once. Nb and %: number and percentage of compounds in each category. R: ratio between the percentage of compounds in each category comprised in the “best-E3” subset resulting from VS and the total percentage of compounds in each category given as input to VS. R values > 1 indicate that there has been an enrichment of compounds in a given category in the “best-E3” subset with respect to the total percentage of compounds in the same category used for VS. R values < 1 indicate that compounds in a given category are under-represented within the “best-E3” subset.

Category	Compounds Used for VS		“Best-E3”								
			5HK1			6DK1			ERG2		
	Nb	%	Nb	%	R	Nb	%	R	Nb	%	R
**Ago-Ant**	36	0.1	8	0.4	2.8	5	0.1	1.0	10	0.3	2.4
**Metab**	15,871	58.1	433	20.2	0.3	694	19.0	0.3	1384	43.3	0.7
**Ste_Lip**	3761	13.8	436	20.4	1.5	1192	32.7	2.4	627	19.6	1.4
**Sterols**	1593	5.8	412	19.2	3.3	706	19.3	3.3	299	9.3	1.6
**Steroids**	362	1.3	109	5.1	3.8	138	3.8	2.9	218	6.8	5.1
**Androg**	101	0.4	52	2.4	6.6	68	1.9	5.0	79	2.5	6.7
**Estrog**	63	0.2	42	2.0	8.5	6	0.2	0.7	45	1.4	6.1
**Chol_P**	442	1.6	135	6.3	3.9	254	7.0	4.3	140	4.4	2.7
**Ergo_P**	347	1.3	124	5.8	4.6	169	4.6	3.6	85	2.7	2.1
**FDA**	1538	5.6	142	6.6	1.2	157	4.3	0.8	269	8.4	1.5
**World**	3192	11.7	248	11.6	1.0	261	7.2	0.6	43	1.3	0.1
**All Ste**	6669	24	1310	61	2.5	2533	69	2.8	1493	47	1.9
**All Div**	20,601	75	823	38	0.5	1112	30	0.4	1696	53	0.7
**All Cat**	27,306	100	2141	100	1	3650	100	1	3199	100	1

**Table 3 ijms-24-06367-t003:** Summary of the results of VS experiments against the 3D structures of *Xl*σ1-R in coordinate files 7W2B and 7W2E. 7W2E_A and 7W2B_A: monomer with chain ID “A” in coordinate files 7W2E and 7W2B, respectively. B_20, E_3.0, Nb. Hits, E_calc_ (kcal/mol), Hb, Contacts and Clashes are as defined in Table 1.

	7W2E_A		7W2B_A	
**Subset**	B_20	E_3.0	B_20	E_3.0
**Nb. Hits**	20	143	20	90
**E_calc_ (kcal/mol)**	−10.8–(−9.2)	−10.8–(−7.8)	−11.1–(−9.3)	−11.1–(−8.1)
**Hb**	0–5	0–8	0–6	0–8
**Contacts**	31–93	18–100	37–110	22–110
**Clashes**	0–3	0–6	0–5	0–5

**Table 4 ijms-24-06367-t004:** Average B-factor values of the five selected molecules after fitting in the electron density map of the *Xl*σ1-R structure in chain C of coordinate files 7W2E (7W2E_C) and 7W2B (7W2B_C). For each compound in each structure, occupancy = 1.00.

YMDB ID	Ligand Name	Average B-Factor with 7W2E_C	Average B-Factor with 7W2B_C
YMDB00543	Ergosterol	86.4	114.7
YMDB01653	Catechin	107.2	131.8
YMDB00293	7,8-Dihydropteroic acid	126.2	121.1
YMDB01754	Myricetin	116.1	108.3
YMDB00452	3′,5′-Cyclic dAMP	163.8	139.0

**Table 5 ijms-24-06367-t005:** Average B-factor values of all monomers of the *Xl*σ1-R structures in coordinate files 7W2E (apo form, open conformation) and 7W2B (apo form, closed conformation).

7W2E			7W2B		
Chain ID	Number of Atoms	Average B-Factor	Chain ID	Number of Atoms	Average B-Factor
A	1754	92.8	A	1701	64.9
B	1754	105.3	B	1729	70.2
C	1754	104.8	C	1701	81.4
D	1754	97.4	D	1742	73.8
E	1741	114.5	E	1701	71.0
F	1754	111.0	F	1701	87.8
G	1748	92.2	G	1729	86.0
H	1726	100.1	H	1701	101.8
I	1754	100.4	I	1714	109.2
J	1754	90.0	J	1657	101.9
K	1718	103.0	K	1692	99.9
L	1748	104.7	L	1701	82.9

**Table 6 ijms-24-06367-t006:** Dissociation constant (K_D_) values determined by fluorescence titration.

Ligands	K_D1_ (μM)	K_D2_ (μM)
Pridopidine	0.254 ± 0.122	177 ± 67.1
Iloperidone	0.019 ± 0.015	33.10 ± 14.41
16,17-didehydroprogesterone	0.010 ± 0.004	15.81 ± 4.03

## Data Availability

The data presented in this study are available on request from the corresponding author.

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
