# Peer review of "Investigation of the Entry Pathway and Molecular Nature of σ1 Receptor Ligands"

_ijms, 2023, doi:10.3390/ijms24076367_

Round 1

Reviewer 1 Report

In the manuscript “Investigation of the entry pathway and molecular nature of σ1 receptor ligands”, authors simulated conventional molecular dynamic (MD) to decide which of two proposed ligand-entry pathways is more likely for ligands to enter the binding site, siding for the entrance between α-helices adjacent to the membrane. In addition, authors performed virtual screening (VS) to tip potential binders and verified their binding using “fluorescent titration” of intrinsic protein fluorescence. These two parts are interconnected very weakly. I do not know whether the connection between these two parts can be improved or manuscript should be rather split in two.

On MD part

1. Author modelled trimeric σ1 receptor. σ1 receptor is a membrane protein. X-ray structures used for modelling do not contain membranes. The stereo angle between the N-terminal α-helix and globular domain varies among protomers. The output of MD strongly varies among protomers as well. How many MD replicas were simulated? Several replicas are needed to find out whether variation among protomers is just a matter of particular MD (e.g. variations among replicas are of the same magnitude as variation among protomers) of system setup (e.g. variations among protomers are greater than among replicas).

2. Description of how the system was constructed, especially how the trimer was immersed into the membrane is missing.

3. Membrane patch is quite large and almost protein free. How the membrane was the membrane stabilised?

4. How does 1.5 µs of conventional MD compare to hundreds (e.g., 500) of ns of accelerated MD?

5. The authors speculate on possible entry pathways based on conformational changes of the apo form of the receptor. Would not it be more straightforward to simulate the dissociation of the ligand-receptor complex?

On VS particular

6. Were decoys or at least negative controls used to verify VS.

Ln 951 - 954 The authors write “In spite of the high overall similarity between the two Hsσ1-R structures (the RMSD values calculated after optimal superposition of residues in the C-term ligand-binding domain are in the range 0.4-0.6 Å for Cα atoms and 0.6-0.9 Å for all atoms) and between the two apo Xlσ1-R structures (Supplementary Table 3), there was no correlation between the Ecalc values between the “best-E3” hits of 5HK1 and 6DK1 or those of 7W2B and 7W2E”. Commonly, a subtle difference in receptor structure may considerably affect energy contributions to the binding pose. This could be overcome by energy minimization of the VS poses confined closely to the original ligand pose followed by VINA local search re-scoring.

Other points

Ln 446 - Eq. 2 is not the Hill equation. It is the sum of two rectangular hyperbolas describing saturation binding to two independent binding sites. How this fits the idea of trimers (= 3 presumably cooperative sites).

Reviewer 2 Report

This manuscript by Pascarella was set to further understand the entry pathway of transmembrane xlσ-R ligands, by molecular dynamic simulations, virtual screen, electron density map fitting and ligand binding test. The written style is clear, and experimental methods are well described. My major critics is that, While the rout of ligand access has being perform using MD simulation and experimental in multiple studies with very similar results as in this story.( PMID: 30291362, and https://doi.org/10.1021/acs.jcim.9b00649 and others), the authors should provide and discuss more what new discoveries have been found in the experiments, to convince readers that this story is novel.

Major points:

1. Section 3.4, the authors were trying to do fitting of selecting compounds into xlσ1R PDB electron density. You CAN NOT fit any ligands into the original diffraction files, unless you 100% know the ligands are present in the crystal condition.  The authors should explain how it is performed and why.

Minor points:

Figure 6, experiment should be repeated, with statistic number n, and error bars in the fitting."

Reviewer 3 Report

The paper by Pascarella et al reports a molecular dynamics and experimental study on the transmembrane protein σ1. The work is aimed at identifying the entry path of ligands in the pocket of the σ1 receptor and at defining the possible ligands. The topic is quite relevant, since it is related to a protein that is involved in neurodegenerative diseases and understading the mechanism of ligand binding has potential drug-design applications. Exstensive molecular dynamics simulations are performed starting from 3D structures coming from the pdb data bank and from the alphafold database, including the proteins together with a lipid bilayer. All the MD simulations details are provided and the procedure is clearly written. The results appear to be robust, giving information of previously not fully explored interaction between the σ1 receptor, its homologous ERG2 and the ligands. To complement the computational results, the authors performed fluorescence titration measurements to estimate the ligand binding, thus providing an experimental validation for the in silico results. The paper is clearly written, the reference to the literature is adequate and the conclusions are properly supported by the results. The paper is also of interest to the journal’s readers community and therefore I can endorse it for publication on the “International Journal of Molecular Science” in its current form.

Round 2

Reviewer 1 Report

The authors fully addressed all my points except the need for several replicas of MD. The authors correctly point out that "several MD replicas should be employed to get convergence of structural, dynamical, and energetic properties averaged over the phase space" and "More replicas would increase the sampling of the phase space, but they would not change the evidence reported in this work about the ligand-entry pathway identified for one of the three monomers."  However, whether differences among trajectories of individual monomers result from variation in initial structures or assigned initial velocities remain unanswered. If authors for whatever reason (lack of computing resources, lack of curiosity, ...) refrain from adding MD replicas then they should not discuss differences among monomers and focus only on events and features that are common to all monomers in MD trajectories.

Reviewer 2 Report

The authors have addressed my concerns, and I recommend this story for publication. 
